# Effects of Aminomethylphosphonic Acid on the Transcriptome and Metabolome of Red Swamp Crayfish, *Procambarus clarkii*

**DOI:** 10.3390/ijms25020943

**Published:** 2024-01-11

**Authors:** Tao Mao, Jinhua Gan, Keping Yuan, Li He, Yali Yu, Ziduo Liu, Yuntao Zhou, Gaobing Wu

**Affiliations:** 1State Key Laboratory of Agricultural Microbiology, College of Life Science and Technology, Huazhong Agricultural University, Wuhan 430070, China; mtao327@163.com (T.M.); lzd@mail.hzau.edu.cn (Z.L.); 2Yangtze River Fisheries Research Institute, Chinese Academy of Fishery Sciences, Wuhan 430223, China; gjh@yfi.ac.cn (J.G.); ykp@yfi.ac.cn (K.Y.); heli28@sohu.com (L.H.); ylyu8811@yfi.ac.cn (Y.Y.)

**Keywords:** AMAP, *P. clarkii*, transcriptomics, metabolome

## Abstract

Red swamp crayfish, *Procambarus clarkii* (*P. clarkii*), is an important model crustacean organism used in many types of research. However, the effects of different doses of aminomethylphosphonic acid (AMAP) on the transcriptome and metabolites of *P. clarkii* have not been explored. Thus, this study investigated the molecular and metabolic mechanisms activated at the different exposure dosages of AMAP in *P. clarkii* to provide new insights into the strategies of *P. clarkii* in response to the high concentrations of AMAP in the environment. In the present study, the *P. clarkii* were divided into three groups (control group; low-dosage AMAP exposure; high-dosage AMAP exposure), and hepatopancreatic tissue samples were dependently taken from the three groups. The response mechanisms at the different dosages of AMAP were investigated based on the transcriptome and metabolome data of *P. clarkii*. Differentially expressed genes and differentially abundant metabolites were identified in the distinct AMAP dosage exposure groups. The genes related to ribosome cell components were significantly up-regulated, suggesting that ribosomes play an essential role in responding to AMAP stress. The metabolite taurine, involved in the taurine and hypotaurine metabolism pathway, was significantly down-regulated. *P. clarkii* may provide feedback to counteract different dosages of AMAP via the upregulation of ribosome-related genes and multiple metabolic pathways. These key genes and metabolites play an important role in the response to AMAP stress to better prepare for survival in high AMAP concentrations.

## 1. Introduction

In recent years, there has been growing concern regarding the impact of aminomethylphosphonic acid (AMAP) (N-phosphonomethylglycine) on aquatic species [1,2]. Glyphosate is the most widely used non-selective herbicide for controlling weed growth worldwide [3,4], and its natural degradation product was AMPA. In soils and surface waters, AMPA can undergo a process of degradation mediated by microbial communities, during which AMPA is formed, resulting in the main breakdown product [5,6,7]. Generally, high levels of AMPA and AMPA in surface water occur with the first runoff episode after application [8].

The red swamp crayfish, *Procambarus clarkii* (*P. clarkii*), originally from America, was introduced into China and has multiplied and proliferated abundantly in the years that followed [9]. Crayfish farming is becoming a developing aquaculture industry in China, especially in Jiangsu Province, with a gross output value of more than USD 200 millionin 2009 [10]. The biological response of *P. clarkii* to exposure to AMPA involves changes at the biochemical and cellular levels, which in turn cause changes in the structure and function of cells and tissues and, ultimately, changes in the physiology and behavior of organisms [11,12]. Therefore, the potential impact of exposure to AMPA is important in the assessment of *P. clarkii* as a commercial product. Several transcriptome and metabolome studies of *P. clarkii* have been reported in recent years [13,14,15]. Studies have suggested that virus infection might affect the growth of *P. clarkii* via the use of high-throughput transcriptome sequencing [16]. Based on a targeted and untargeted metabolomic analysis, Moro et al. investigated the metabolic impairments caused by diclofenac in *P. clarkii* [17,18]. However, limited information is available about the effects of exposure to AMPA at different dosages when aiming to assess the effects on the transcriptome and metabolome. 

There are limited data on the effects of AMPA on freshwater crayfish, in particular *P. clarkii*. The objective of this study was to investigate the effects of exposure to different dosages of AMPA on the gene expression and metabolite abundance in *P. clarkii*. A control, low (100 ppb) dosage, and high (1000 ppb) dosage exposure to realistic concentrations of AMPA were performed, and the effects on the transcriptome and metabolome were sequenced. Differentially expressed genes (DEGs) and differentially abundant metabolites were identified in distinct AMPA dosage groups. GO (Gene Ontology) and KEGG (Kyoto Encyclopedia of Genes and Genomes) functional analyses were performed to explore the impact of AMPA on the molecular and metabolites in *P. clarkii*. 

## 2. Results

### 2.1. Identification of AMPA Dosage-Related DEGs in P. clarkii

Our study identified DEGs related to low (100 ppb) dosage and high (1000 ppb) dosage AMPA exposure. In total, 1109 significantly up-regulated genes and 611 significantly down-regulated genes were selected between the low-dosage AMPA-exposed *P. clarkii* and control samples (Figure 1A). In total, 1639 significantly up-regulated genes and 844 significantly down-regulated genes were selected between the high-dosage AMPA-exposed *P. clarkii* samples and control samples. (Figure 1B). Between the high-dosage and low-dosage AMPA exposure in *P. clarkii*, we identified 288 over-expressed and 202 down-expressed genes (Figure 1C). In addition, 64 DEGs were identified simultaneously in the three groups (Figure 1D). The overlapping of DEGs included LOC123767151, LOC123774039, and LOC123774908. Real-time quantitative PCR also found that LOC123767151, LOC123774039, and LOC123774908 were up-regulated in *P. clarkii* with low-dosage and high-dosage AMPA exposure compared with the control *P. clarkii* samples (Figure 1E–G). LOC123767151, LOC123774039, and LOC123774908 were annotated in multiple “Biological Process” terms, including prolyl-tRNA aminoacylation, tRNA aminoacylation for protein translation, the regulation of transcription, and the nucleoside metabolic process (Figure 2). 

Moreover, we annotated the DEGs using the GO term to investigate the function of the DEGs. The DEGs from the three groups simultaneously annotated the cellular process, metabolic process, single-organism process terms in “Biological Process”, the catalytic activity, signal transducer activity, and binding terms in “Molecular Function”, the membrane, macromolecular complex, and organelle part terms in “Cellular Component”, and so on (Figure 2A–C). Thus, these results implied that high-dosage and low-dosage AMPA-related DEGs were involved in the same biological terms in *P. clarkii*.

### 2.2. Functional Enrichment Analysis of AMPA Doses Related DEGs 

We performed GO and KEGG pathway enrichment analysis based on the detected DEGs from the three groups. The DEGs between the low-dosage AMPA and control samples were involved in ribosome-related function, including the structural constituents of the ribosome, small ribosomal subunit, and ribosome GO terms (Figure 3A). The DEGs between the high-dosage AMPA and control samples were also enriched for protein peptidyl-prolyl isomerization and mitochondrial inner membrane GO terms, in addition to ribosome-related GO terms (Figure 3B). Additionally, the DEGs between the high-dosage and low-dosage AMPA samples were enriched in oxidoreductase activity, the extracellular region, lipid transporter activity, and lipid transport GO terms (Figure 3C). KEGG pathway enrichment analysis suggested that the DEGs from the low-dosage and high-dosage AMPA were simultaneously enriched in the thermogenesis, retrograde endocannabinoid signaling, and oxidative phosphorylation pathways (Figure 3D,E). Moreover, the DEGs between the high-dosage and low-dosage AMPA samples were related to multiple biosynthetic and metabolic pathways, such as steroid biosynthesis, arginine biosynthesis, pyrimidine metabolism, and amino sugar and nucleotide sugar metabolism (Figure 3F). These results suggest that different doses of AMPA may affect the metabolic functions of *P. clarkii*.

### 2.3. Identifying AMPA Dosage-Related Metabolites in P. clarkii

As the pathway serves as an enrichment function for DEGs, we assessed the AMPA dosage of different metabolites. This work identified different metabolites between the low-dosage AMPA and control *P. clarkii* samples, between the high-dosage AMPA and control *P. clarkii* samples, and between the high-dosage AMPA and low-dosage AMPA *P. clarkii* samples (Figure 4A). The number and levels of metabolites between the three groups are presented in Figure 4B–D. This work selected 15 different metabolites between the low-dosage AMPA and control *P. clarkii* samples, including seven different high-abundance metabolites and eight different low-abundance metabolites (Figure 4E). Twelve different metabolites were identified between the high-dosage AMPA and control *P. clarkii* samples, including ten different high-abundance metabolites and two different low-abundance metabolites (Figure 4F). In addition, we identified seven different metabolites between the high-dosage AMPA and high-dosage AMPA exposure *P. clarkii* samples, including five different high-abundance metabolites and two different low-abundance metabolites (Figure 4G). Furthermore, only one different metabolite was identified simultaneously in the three groups (Figure 4H). In brief, exposure to different dosages of AMPA also impacts the metabolome of *P. clarkii*. 

### 2.4. Functional Enrichment Analysis of AMPA Dosage-Related Differential Metabolites

We next investigated the correlation between different metabolites and the biological properties associated with different metabolites. This study found that different metabolites identified from three groups showed a correlation in their metabolite abundance (Figure 5A–C). In particular, several metabolites showed a positive correlation between the high-dose AMPA and control *P. clarkii* samples. (Figure 5B). The KEGG pathway enrichment analysis suggested that different metabolites from three groups were enriched in some of the same pathways, such as carbon metabolism, taurine and hypotaurine metabolism, ABC transporters, and cysteine and methionine metabolism pathways (Figure 5D–F). Overall, these findings demonstrate that the AMPA dosage correlated with the metabolite abundance and metabolic function in *P. clarkii*.

## 3. Discussion

There were few studies on the effects of AMPA exposure on Procrazii clarkii, especially in combination with transcriptomes and metabolomes. The present study found that DEGs and different metabolites were associated with different dosages of AMPA in the *P. clarkii* model. Some DEGs were validated by real-time quantitative PCR. The genes related to ribosome cell components were up-regulated, suggesting that the ribosome played an essential role in responding to AMAP stress. This work suggested that the dosage of AMPA influences metabolic pathways and functions. This is the first study to simultaneously compare the effects of different doses of AMPA on *P. clarkii* metabolites and transcriptomes. 

The functional enrichment analysis of DEGs suggested that high-dosage exposure to AMPA affected the lipid transporter activity and lipid transport function of *P. Clarkii* (Figure 3C). A recent study found that AMPA exposure induced lipid metabolism disorder in grass carp based on metabolomics analysis [19]. Meanwhile, Sulukan et al. reported that lipid accumulation was detected in a zebrafish model exposed to AMPA [20]. Therefore, the present work concluded that exposure to AMPA may affect the growth and function of aquatic organisms by affecting lipid metabolism [21,22,23]. The capability of lipopolysaccharide to influence ribosome function has already been demonstrated in *P. Clarkii* [24]. In the present study, we discovered that ribosome-associated functions were up-regulated in both the high- and low-dose AMPA groups, suggesting that AMAP exposure also affects ribosome functions in *P. Clarkii* (Figure 3A,B,D,E). Yan et al.’s study provides novel insights into the effects of glyphosate on fish species by regulating metabolites, such as bile acids and short-chain fatty acids [19]. This work selected different metabolites from the high-throughput quantification for metabolites and performed a functional enrichment analysis of the different metabolites. We found that the metabolism pathways were identified in the three AMPA groups. This result is consistent with previous reports finding that AMPA affects the metabolism disorders of aquatic organisms [25,26]. 

Although our work was limited by a large number of samples, we used three AMPA exposure groups of *P. clarkii* samples to analyze the influence of AMPA on both the transcriptome and metabolome. In the future, we will expand the samples and add different dosages of AMPA exposure to comprehensively explore the mechanisms implicated in the effect of AMPA on metabolic function. In addition, the exposure of *P. clarkii* to high dosages of AMPA showed alterations in several metabolic pathways, which warrants future detailed biological experiments to validate these discoveries.

## 4. Materials and Methods

### 4.1. Ethics Statement 

This study was approved by the Animal Care and Use Committee of Yangtze River Fisheries Research Institute (Permit No. 2022yfimaotao001). 

### 4.2. P. clarkii Exposure to AMPA

The red crayfish (weight, 30 ± 5 g) were purchased from a farm in Hubei Province, China, and were temporarily bred at a water (DO > 5 mg/L, ammonium nitrogen < 0.2 mg/L, pH 6~9) temperature of 25 ± 1 °C for 7 days without testing for any abnormalities. In total, 120 healthy red swamp crayfish, half male and half female, were randomly selected and divided into three groups, with 40 in each group. Two groups were prepared for breeding in low (100 ppb)-dosage and high (1000 ppb)-dosage exposures of AMPA. The LC50 of the AMPA was 124 mg/L. The rest of the *P. clarkii* samples were used as a control group. The AMPA was purchased from Sigma-Aldrich as a 40% water solution of N-phosphonomethyl glycine-monoisopropylamine salt, was diluted in phosphate-buffered saline (PBS) 1X, and adjusted to pH 7.4 with NaOH [27,28]. AMPA with a purity above 98% (analytical standard) was obtained from Sigma-Aldrich Chemical Company (Spain, Madrid). 

### 4.3. RNA Extraction, Transcriptome Library Construction and Sequencing

RNA was isolated from *P. clarkii* hepatopancreatic tissue and RNA-sequencing performed. The total RNA was isolated with Oligo(dT). RNA-seq libraries were constructed according to the manufacturer’s protocol. The mRNA was divided into fragments using a fragmentation buffer. The first cDNA strand was synthesized using an mRNA template with six-base random hexamers. A second strand of cDNA was then synthesized by adding buffer, dNTPs, RNase H, and DNA polymerase I. After purification with the QiaQuick PCR kit and elution with EB buffer solution, the terminal repair was performed, poly (A) was added, and the sequencing bridge was connected. The fragment sizes were selected via agarose gel electrophoresis and PCR amplification was performed. Then, the quality-checked libraries were sequenced on the Illumina HiSeq™ platform. RNA reads were aligned to the reference genome assembly using hisat2 (version 2.2.1) software. Then, we used FastQC software (version 0.11.9, http://www.bioinformatics.babraham.ac.uk/projects/fastqc, accessed on 16 March 2022) to control the quality for the subsequent analysis.

### 4.4. Chemicals and Reagents

LC–MS-grade methanol (MeOH) was purchased from Fisher Scientific (Loughborough, UK). 2-amino-3-(2-chlorophenyl)-propionic acid was purchased from Aladdin (Shanghai, China).

### 4.5. Sample Preparation and Standards

For this work, an appropriate amount of *P. clarkii* hepatopancreatic tissue sample was weighted into a 2 mL centrifuge tube, and a 100 mg glass bead was added. Next, 1000 µL of 50% methanol water (stored at 4 °C) was added and vortexed for 30 s. We placed the centrifuge tube containing the sample in the 2 mL adapter matched to the instrument, immersed it in liquid nitrogen for rapid freezing for 5 min, removed the centrifuge tube and thawed it at room temperature, placed the centrifuge tube back in the 2 mL adapter, placed it in the tissue grinder, and ground it at 55 Hz for 2 min. We repeated the last step twice. The centrifuge tube was removed; after being centrifuged for 10 min at 12,000 rpm and 4 °C, the entire supernatant was removed, transferred to a new 2 mL centrifuge tube, concentrated, and dried. The sample was then redissolved by adding 300 μL of 2-amino-3-(2-chlorophenyl)-propionic acid (4 ppm) solution prepared with 50% methanol-water (stored at 4 °C) to redissolve the sample, and the supernatant was filtered through a 0.22 μm membrane and transferred to the detection bottle for LC–MS detection.

### 4.6. Differential Expression Genes and Differential Metabolite Abundance in Three Dose Groups of AMPA

The “edgeR” package of the R 4.0.3 software was used to analyze the differential expression genes (DEGs) in the three low-dosage groups, including three low-dosage samples versus three control *P. clarkii*, three high-dosage versus three control *P. clarkii*, and three high-dosage versus three low-dosage *P. clarkii* samples. After significance analysis and false discovery rate (FDR) adjustment, FDR < 0.05 and fold change (FC) > 2 were considered as significantly up-regulated genes, and FDR < 0.05 and FC < −2 were considered as significantly down-regulated genes.

### 4.7. Real-Time Quantitative PCR Quantify Differential Expression Genes

The total RNA of the liver tissues was isolated using OHAUS-5515R following the manufacturer’s instructions (OHAUS, Parsippany-Troy Hills, NJ, USA). The RNA concentration was determined using NanoDrop 8000 (Thermo Fisher, Waltham, MA, USA). The expression levels of mRNAs were determined using the StepOnePlusTM Real-Time System. Based on the gene sequence information, primers for qPCR amplification were designed using the primer design software Primer 5. The relative levels of expression of mRNAs were analyzed using the 2^−ΔΔCt^ method [29,30].

### 4.8. GO Term Annotation and KEGG Pathway Enrichment 

Based on the results of the DEGs, we performed GO enrichment analysis using clusterProfiler (version 4.0.2) software. Significant GO terms and KEGG pathways were identified via a hypergeometric test, and an FDR was used to correct the *p*-value. The statistical significance was set at an adjusted *p*-value < 0.05.

### 4.9. Statistical Analyses

All analyses and graphs were performed and made using R software (v 4.0.3). The “edgeR” package was used to identify the DEGs and a hypergeometric test was used to perform GO term annotation and KEGG pathway enrichment. After FDR adjust, an adjusted *p*-value < 0.05 indicates a statistically significant. 

## 5. Conclusions

Overall, this work explored the effects of different doses of AMAP on the transcriptome and metabolites of *P. clarkii*. Differentially expressed genes and differentially abundant metabolites were identified in the distinct AMAP dosage groups. The genes related to ribosome cell components were significantly up-regulated, suggesting that ribosomes play an essential role in responding to AMAP stress. *P. clarkii* may provide feedback to counteract different dosages of AMAP via the upregulation of ribosome-related genes and multiple metabolic pathways.

## Figures and Tables

**Figure 1 ijms-25-00943-f001:**
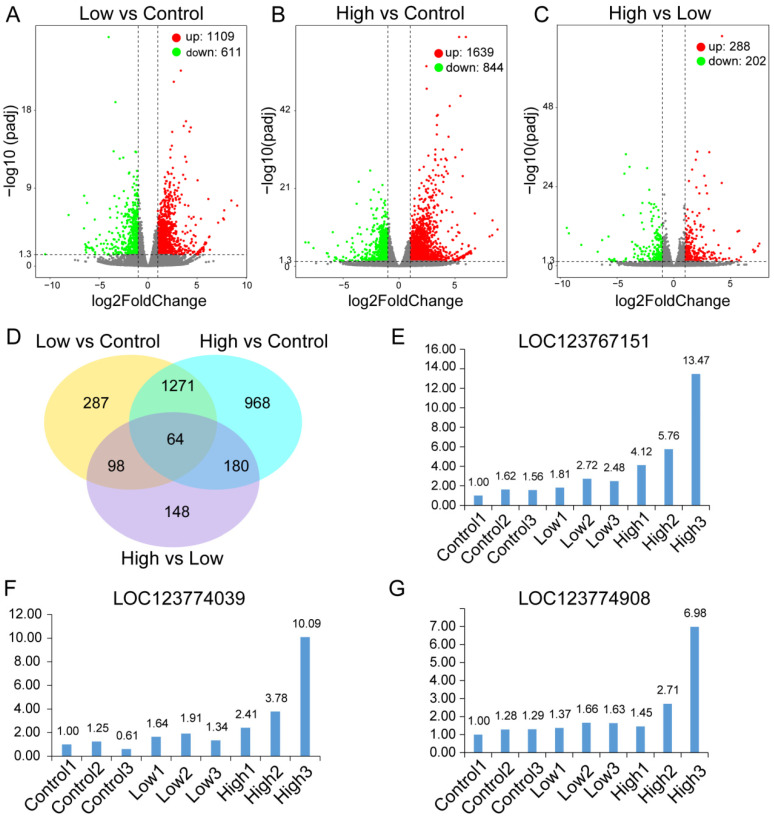
Identification of differential expression genes (DEGs) in *P. clarkii* samples. (**A**) Volcano plot exhibiting 1109 up-regulated DEGs and 611 down-regulated DEGs between low-dosage AMPA and control *P. clarkii* samples. (**B**) Volcano plot exhibiting 1639 up-regulated DEGs and 844 down-regulated DEGs between high-dosage AMPA and control *P. clarkii* samples. (**C**) Volcano plot exhibiting 288 up-regulated DEGs and 208 down-regulated DEGs between high-dosage AMPA and low-dosage AMPA *P. clarkii* samples. (**D**) Venn diagram represents the overlapping of DEGs from low-dosage AMPA and control, high-dosage AMPA and control, and high-dosage AMPA and low-dosage AMPA *P. clarkii* samples. (**E**) Expression levels of LOC123767151 estimated from qPCR in control, low-dosage, and high-dosage AMPA-exposed samples. (**F**) Expression levels of LOC123774039 estimated from qPCR in control, low-dosage, and high-dosage AMPA-exposed samples. (**G**) Expression levels of LOC123774908 estimated from qPCR in control, low-dosage, and high-dosage AMPA-exposed samples. DEGs, differentially expressed genes.

**Figure 2 ijms-25-00943-f002:**
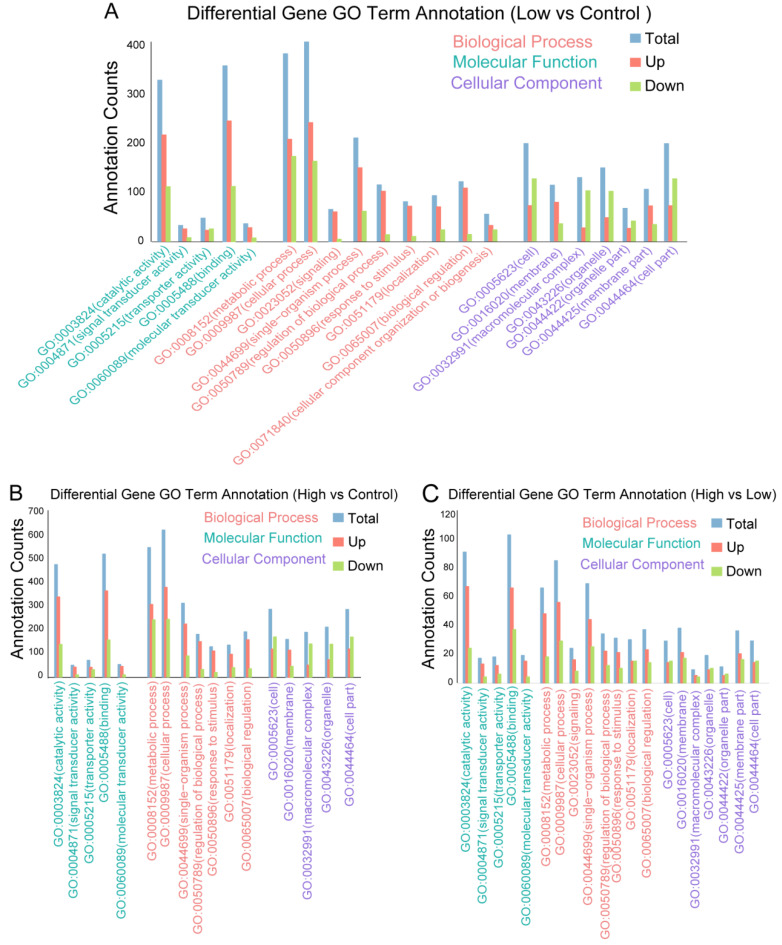
Annotation number of DEGs. (**A**) The number of GO annotations of DEGs between low-dose AMPA and control *P. clarkii* samples. (**B**) The number of GO annotations of DEGs between high-dose AMPA and control *P. clarkii* samples. (**C**) The number of GO annotations of DEGs between high-dose AMPA and low-dose AMPA *P. clarkii* samples. DEGs, differentially expressed genes; GO, gene ontology.

**Figure 3 ijms-25-00943-f003:**
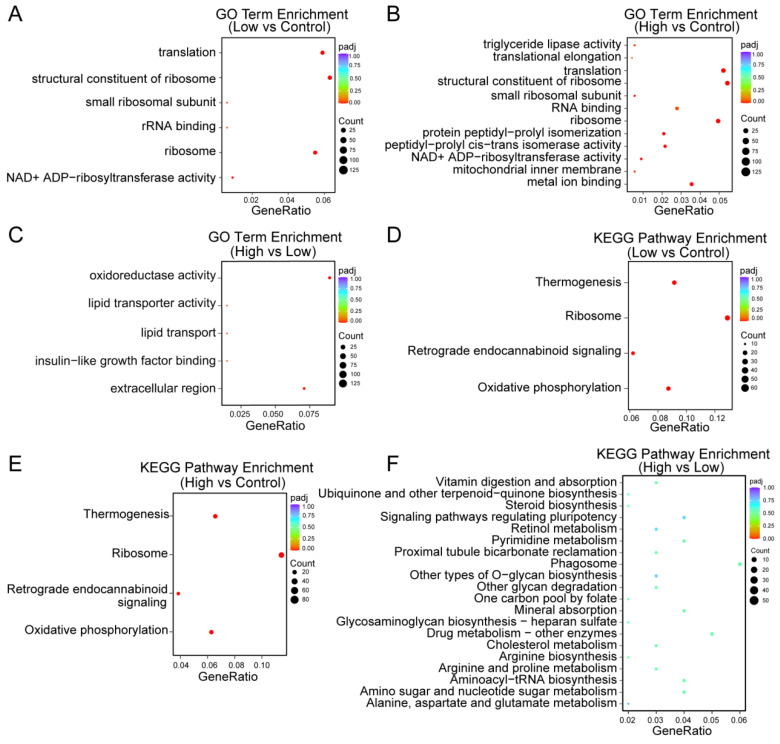
Functional analysis for DEGs. (**A**) GO categories of DEGs between low-dose AMPA and control *P. clarkii* samples. (**B**) GO categories of DEGs between high-dose AMPA and control *P. clarkii* samples. (**C**) GO categories of DEGs between high-dose AMPA and low-dose AMPA *P*. *clarkii* samples. (**D**) KEGG pathway enrichment analysis of DEGs between low-dose AMPA and control *P. clarkii* samples. (**E**) KEGG pathway enrichment analysis of DEGs between high-dose AMPA and control *P. clarkii* samples. (**F**) KEGG pathway enrichment analysis of DEGs between high-dose AMPA and low-dose AMPA *P. clarkii* samples.

**Figure 4 ijms-25-00943-f004:**
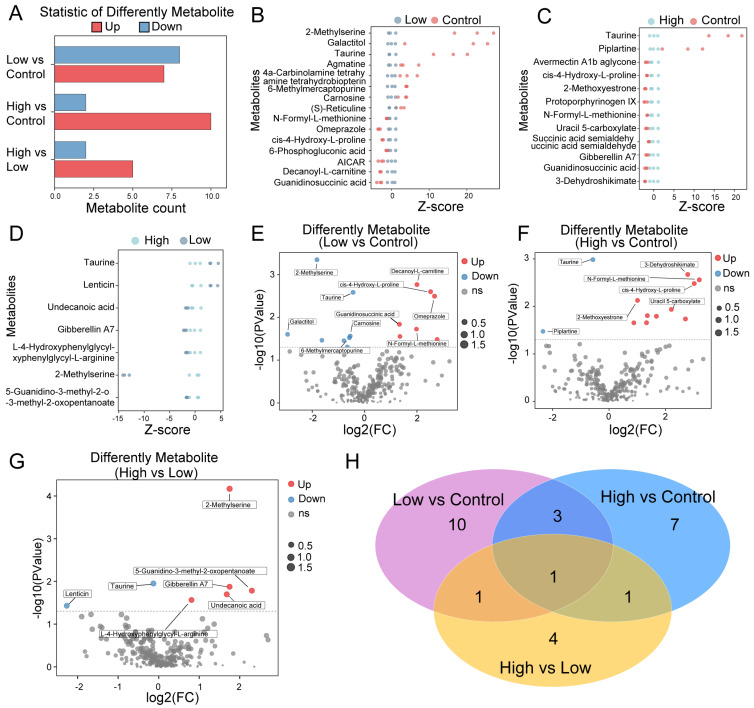
Identification of different metabolites in *P. clarkii* samples. (**A**) Statistics of differently up-regulated and down-regulated metabolites between low-dosage AMPA and control, high-dosage AMPA and control, and high-dosage AMPA and low-dosage AMPA *P. clarkii* samples. (**B**) Metabolite abundance in low-dosage AMPA and control *P. clarkii* samples. (**C**) Abundance of metabolites in high-dosage AMPA and control *P. clarkii* samples. (**D**) Abundance of metabolites in high-dosage AMPA and low-dosage AMPA *P. clarkii* samples. (**E**) Volcano plot exhibiting seven up-regulated metabolites and eight down-regulated metabolites between low-dose AMPA and control *P. clarkii* sample. (**F**) Volcano plot showing ten up-regulated metabolites and two down-regulated metabolites between high-dose AMPA and control *P. clarkii* sample. (**G**) Volcano plot showing five up-regulated metabolites and two down-regulated metabolites between high-dosage AMPA and low-dosage AMPA *P. clarkii* sample. (**H**) Venn diagram represents the overlapping of the different metabolites from three groups.

**Figure 5 ijms-25-00943-f005:**
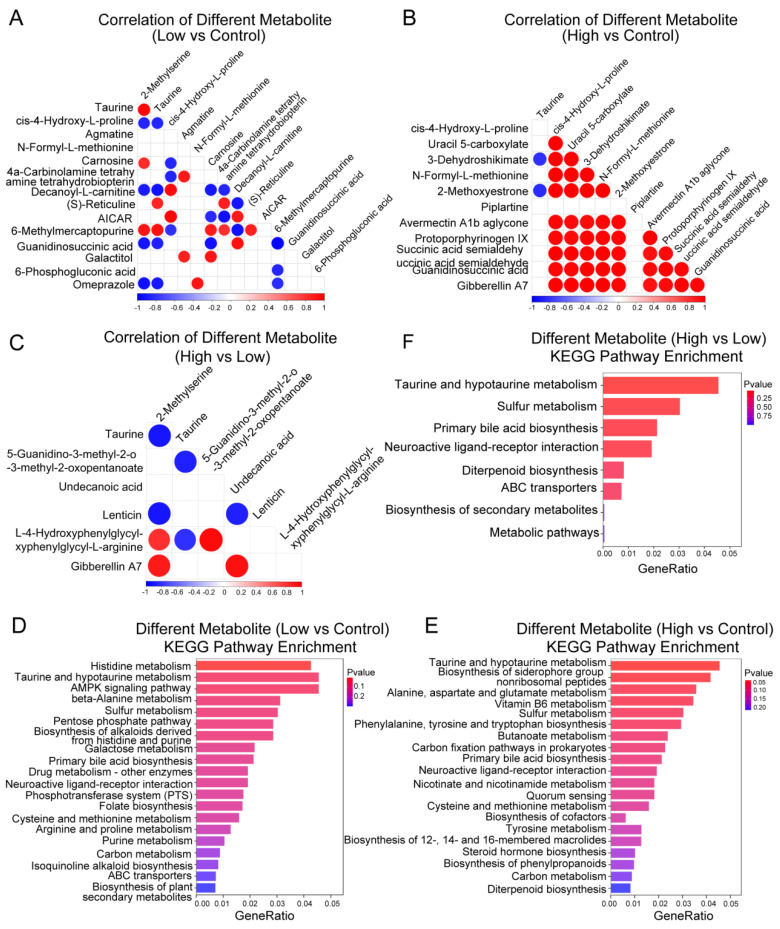
Correlation analysis and functional analysis for different metabolites. (**A**) Correlation analysis of different metabolites between low-dosage AMPA and control *P. clarkii* samples. (**B**) Correlation analysis of different metabolites between high-dosage AMPA and control *P. clarkii* samples. (**C**) Correlation analysis of different metabolites between high-dosage AMPA and low-dosage AMPA *P. clarkii* samples. (**D**) KEGG pathway enrichment analysis of different metabolites between low-dose AMPA and control *P. clarkii* sample. (**E**) KEGG pathway enrichment analysis of different metabolites between high-dose AMPA and control *P. clarkii* sample. (**F**) KEGG pathway enrichment analysis of different metabolites between high-dose AMPA and low-dose AMPA *P. clarkii* sample.

## Data Availability

The data used for the analysis is available upon request.

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
