# Peer review of "Effects of Aminomethylphosphonic Acid on the Transcriptome and Metabolome of Red Swamp Crayfish, *Procambarus clarkii"

_ijms, 2024, doi:10.3390/ijms25020943_

Round 1

Reviewer 1 Report

Comments and Suggestions for Authors

The Decision on the manuscript with ID (ijms-2796632-peer-review-v1) is “Major Revisions”. The authors should prepare a point-by-point response to the comments and points raised by the reviewer.

General questions: -

§  Q1. The main problem in this manuscript is that it needs extensive English Editing and proofreading.

§  Q2. The name of the red swamp crayfish (Procambarus clarkii) is mentioned and written 70 times throughout the whole manuscript excluding references. You should write it abbreviated as P. clarkii italic after its first appearance in the text and not repeat it in any sections of the manuscript.

§  Q3. In the reference section, you should write Latin names in italics.

§  Q4. Add Red Swamp Crayfish in the title.

§  Q5. Why the authors did not measure metabolites in the hepatopancreatic tissue homogenate or serum samples before using transcriptomic analysis of associated genes. This is a simple direct method and will support your findings. I suggest adding these results if samples are still present.

Specific questions: -

My specific questions will be focused principally on the Material and Methods and Discussion sections.

§  Q1. Line 223: You should add the water quality parameters (means ±SE) that are present throughout the experiment such as DO, NO2, NH3, and salinity.

§  Q2. Line 226: Why did the authors select these exposure doses? Did you perform LC50 testing before the exposure?

§  Q3. Lines 228-230: Add an appropriate reference for the methodology used to prepare the exposure solution of AMPA.

§  Q4. Line 279: Add an appropriate reference for the methodology used.

§  Livak, K. J., & Schmittgen, T. D. (2001). Analysis of relative gene expression data using real-time quantitative PCR and the 2− ΔΔCT method. Methods, 25(4), 402-408.

§  Q5. Line 287: Write in detail the methods and tests used in statistical analysis.

§  Q6. In the Discussion section, this part needs extensive Editing and reviewing. The authors should add references to support their findings. In addition, the proposed modes of action of AMPA to induce the results should be hypothesized according to your findings. This section should be reformulated again.

Comments on the Quality of English Language

Extensive editing of English language required

Author Response

Response to the comments of Reviewer #1

General questions: -

  • Q1. The main problem in this manuscript is that it needs extensive English Editing and proofreading.

Reply: Thanks for your insightful suggestions. We would like to express our sincere appreciation for your help with our manuscript. We have carefully checked the syntax mistakes in the whole manuscript and made corresponding corrections. We have now worked on both language and readability and have also involved native English speakers for language corrections. We really hope that the flow and language level have been substantially improved. The language of our manuscript has been revised by all authors.

  • Q2. The name of the red swamp crayfish (Procambarus clarkii) is mentioned and written 70 times throughout the whole manuscript excluding references. You should write it abbreviated as P. clarkiiitalic after its first appearance in the text and not repeat it in any sections of the manuscript.

Reply: Thanks for your insightful suggestions. We described Procambarus clarkii as P. clarkii italic after Procambarus clarkii first appearance and did not repeat Procambarus clarkii in any sections of the revised manuscript.

  • Q3. In the reference section, you should write Latin names in italics.

Reply: Thanks for your suggestions. We wrote Latin names in italics in the reference section in the revised manuscript.

  • Q4. Add Red Swamp Crayfish in the title.

Reply: Thanks for your comments. We added “Red Swamp Crayfish” in the title and modified the title as “Effects of aminomethylphosphonic acid on the transcriptome and metabolome of red swamp crayfish, Procambarus clarkii.

  • Q5. Why the authors did not measure metabolites in the hepatopancreatic tissue homogenate or serum samples before using transcriptomic analysis of associated genes. This is a simple direct method and will support your findings. I suggest adding these results if samples are still present.

Reply: Thanks for your comments. We simultaneously measured transcriptome gene expression and metabolites abundance of P. clarkii hepatopancreatic tissue. We added the description in the section of Materials and Methods in the manuscript in line 269-270: “For this work, an appropriate amount of P. clarkii hepatopancreatic tissue sample was weighted into a 2 mL centrifuge tube, and a 100 mg glass bead was added.”

Specific questions: -

My specific questions will be focused principally on the Material and Methods and Discussion sections.

  • Q1. Line 223: You should add the water quality parameters (means ±SE) that are present throughout the experiment such as DO, NO2, NH3, and salinity.

Reply: Thanks for your insightful suggestions. We added water quality parameters in the revised manuscript in line 237-240: “The red crayfish (weight, 30 ± 5g) were purchased from a farm in Hubei Province, China, and were temporarily breeded at water (DO > 5mg/L, ammonium nitrogen < 0.2mg/L, pH 6~9) temperature of 25 ± 1 °C for 7 days without testing for any abnormalities..

  • Q2. Line 226: Why did the authors select these exposure doses? Did you perform LC50 testing before the exposure?

Thanks for your suggestions. We performed the LC50N testing and discovered that the LC50 of AMPA is 124 mg/L. We added the description of LC50N testing in the section of Materials and Methods in lines 243: The LC50 of AMPA is 124 mg/L.. The main reason for choosing the high and low exposure doses was determined by the residue of AMPA in the environment, which was around 0.1 mg/L. Therefore, 0.1 mg/L was chosen as the low exposure doses, and its more than 10 times as high exposure doses [1].

  • Q3. Lines 228-230: Add an appropriate reference for the methodology used to prepare the exposure solution of AMPA.

Reply: Thanks for your insightful suggestions. We added appropriate references for the exposure solution of AMPA in the revised manuscript in line 248-249: “was diluted in phosphate buffered saline (PBS) 1X, and adjusted to pH 7.4 with NaOH [27; 28].”  

  • Q4. Line 279: Add an appropriate reference for the methodology used.
  • Livak, K. J., & Schmittgen, T. D. (2001). Analysis of relative gene expression data using real-time quantitative PCR and the 2− ΔΔCT method. Methods, 25(4), 402-408.‏

Reply: Thanks for your insightful suggestions. We added appropriate references for the methodology in the revised manuscript in line 297-298: “Relative levels of expression of mRNAs were analyzed by 2−ΔΔCt method [29; 30].” 

  • Q5. Line 287: Write in detail the methods and tests used in statistical analysis.

Reply: Thanks for your comments. We added detail the methods and tests to the section of “Materials and Methods” in the revised manuscript in line 305-308: “edgeR” package was used to identify the DEGs and hypergeometric test was used to perform GO term annotation and KEGG pathway enrichment. After FDR adjust, adjusted P-value < 0.05 indicates a statistically significant.

  • Q6. In the Discussion section, this part needs extensive Editing and reviewing. The authors should add references to support their findings. In addition, the proposed modes of action of AMPA to induce the results should be hypothesized according to your findings. This section should be reformulated again.

Reply: Thanks for your comments. We have re-written the Discussion section in the revised manuscript according to the Reviewers suggestion. Moreover, we added references and discussed the action of AMPA in lines 209: A recent study found that AMPA exposure induced lipid metabolism disorder in grass carp based on metabolomics analysis .. We also interpreted the result of ribosome in lines 214:218: The capability of lipopolysaccharide to influence ribosome function has already been demonstrated in P. Clarkii [24]. In the present study, we discovered that ribosome associated function were up-regulated in both high and low dose AMPA groups, suggesting that AMAP exposure also affects ribosome functions in P. Clarkii (Figure 3A-B, D-E).. In addition, explanation of different metabolites was added to the Discussion section in the revised manuscript in lines 218:222: Yan et al. study provides novel insight for the effects of glyphosate on fish species by regulating metabolites, such as bile acids and short-chain fatty acids . This work selected different metabolites from high throughput quantification for metabolites and performed a functional enrichment analysis of the different metabolites.”.

Reviewer 2 Report

Comments and Suggestions for Authors

The paper titled: “Effects of Aminomethylphosphonic Acid on the Transcriptome and Metabolome of Procambarus clarkii” reports several results induced by Aminomethylphosphonic Acid. The authors, in particular, study the effects on the transcriptome and metabolome, using Procambarus clarkii as an experimental model. The proposed experimental model is of considerable importance as it is capable of activating response mechanisms following exposure to various substances.

Furthermore, studies on the effects induced by substances of anthropogenic origin are of particular value, especially when these substances represent a risk for the environment.

I suggest minor revisions:

·         Authors should report the scientific name of the organism using the italics form.

·         In the histograms shown in fig 1 (E, F and G), I suggest eliminating the horizontal lines. This makes the figure more readable.

·         The same for Fig.2. Furthermore, some writings in figure 2 are undersized (X axes).

·         The entire paper must be reviewed for typographical errors, missing or unnecessary spaces... for example P8L122, add a space before brackets.

·         The authors explain the subdivision of the population to be studied but do not refer to the sex of the specimens. Were they equally represented?

·         Finally, the study highlights the effects of different doses of AMAP on the transcriptome and metabolites. Author report a possible role of ribosome in stress response. This aspect should be better argued.

Author Response

Response to the comments of Reviewer #2

The paper titled: “Effects of Aminomethylphosphonic Acid on the Transcriptome and Metabolome of Procambarus clarkii” reports several results induced by Aminomethylphosphonic Acid. The authors, in particular, study the effects on the transcriptome and metabolome, using Procambarus clarkii as an experimental model. The proposed experimental model is of considerable importance as it is capable of activating response mechanisms following exposure to various substances.

Furthermore, studies on the effects induced by substances of anthropogenic origin are of particular value, especially when these substances represent a risk for the environment.

I suggest minor revisions:

  • Authors should report the scientific name of the organism using the italics form.

Reply: Thanks for your comments. We modified the Procambarus clarkii to P. clarkii italic in the revised manuscript.

  • In the histograms shown in fig 1 (E, F and G), I suggest eliminating the horizontal lines. This makes the figure more readable.

Reply: Thanks for your suggestions. In the revised version, we eliminated the horizontal lines in Fig 1E-G.

  • The same for Fig.2. Furthermore, some writings in figure 2 are undersized (X axes).

Reply: Thanks for your suggestions. In the revised version, we have re-edited the figure to ensure that the X-axes text in figure2 is readable (increased the font size in X axes). In addition, we carefully checked all the figures in this work and increased the font size in all the figures.

  • The entire paper must be reviewed for typographical errors, missing or unnecessary spaces... for example P8L122, add a space before brackets.

Reply: Thanks for your insightful suggestions. In the revised version, we have made typographical corrections throughout the revised manuscript. As response in Reviewer1, we have carefully checked the syntax mistakes in the whole manuscript and made corresponding corrections.

  • The authors explain the subdivision of the population to be studied but do not refer to the sex of the specimens. Were they equally represented?

Reply: Thanks for your suggestions. This study randomly screened specimens of half males and half females. In addition, we added the information of specimens sex in the revised manuscript in line 240-241: “120 healthy red swamp crayfish, half males and half females, were randomly selected and divided into three groups with 40 in each group.”.

  • Finally, the study highlights the effects of different doses of AMAP on the transcriptome and metabolites. Author report a possible role of ribosome in stress response. This aspect should be better argued.

Reply: Thanks for your suggestions. As response to Reviewer1 Q6, we added the discussion of the role of ribosome in lines 214:218: The capability of lipopolysaccharide to influence ribosome function has already been demonstrated in P. Clarkii [24]. In the present study, we discovered that ribosome associated function were up-regulated in both high and low dose AMPA groups, suggesting that AMAP exposure also affects ribosome functions in P. Clarkii (Figure 3A-B, D-E)..

Round 2

Reviewer 1 Report

Comments and Suggestions for Authors

The authors appropriately responded to the points and comments raised by the reviewer.

Comments on the Quality of English Language

Extensive editing of English language required